# Newcomer youth's access to contraception care in Canada: A scoping review of qualitative evidence

Zeba Khan[1], Bimbola Olure[2], Victoria Paller[1], Sarah Munro[2,3]*

1 Department of Obstetrics and Gynaecology, The University of British Columbia, Vancouver, British Columbia, Canada, 2 Centre for Advancing Health Outcomes, The University of British Columbia, Vancouver, British Columbia, Canada, 3 Department of Health Systems and Population Health, School of Public Health, University of Washington, Seattle, Washington, United States of America

* sarahmun@uw.edu

## Abstract

### Objective

Many newcomer youth in Canada experience high rates of unmet sexual and reproductive health (SRH) care needs, including contraception. We conducted a scoping review of qualitative evidence to understand newcomer youth's experiences of accessing contraception care in Canada.

### Design and data sources

We followed the Joanna Briggs Institute guidelines to search five databases (MEDLINE, Embase, CINAHL, PsycINFO and Web of Science).

### Eligibility criteria

Qualitative and mixed methods studies describing newcomer youth experiences of SRH in Canada, published in or after 2010 were included.

### Data extraction and synthesis

We adapted relevant headings for our data charting table from the JBI guidelines and a published scoping review on adolescent health. We reported the findings of our analysis as a narrative summary of the included articles.

### Results

We screened 415 titles and abstracts and retrieved 17 eligible studies for full-text review, of which five were included for analysis. Results highlight that sociocultural influences play an important role in newcomer youth's perspectives and access to contraception care. The language preferences of newcomer youth in the context of

**Data availability statement:** All relevant data are within the manuscript and its Supporting Information files.

**Funding:** This study was funded by a University of British Columbia Affiliated Scholarship [to ZK], a Mitacs Fellowship [IT25851 to ZK], a Vancouver Foundation Systems Change Grant [FOI20-5505 to ZK], the Community and University Engagement Support Fund from the University of British Columbia [EXP_041 to ZK], and a Michael Smith Health Research BC Scholar Award [18270 to SM]. The funders had no role in study design, data collection and analysis, decision to publish, or preparation of the manuscript.

**Competing interests:** ZK is a board member of Options for Sexual Health (charity) and the Director of Free Periods Canada (registered non-profit society).

contraception are unique and differ from the needs of the broader immigrant population seeking general health care. Results also identify formal and peer-based educational SRH interventions acceptable to newcomer youth.

## Conclusion

Literature that qualitatively describes newcomer youth's experiences with contraception is scarce, and existing literatures only mentioned contraception in the context of broader SRH-related issues. Further research that specifically focuses on contraception experiences, involves newcomer youth in the research process and explores the roles of peers in contraception access of newcomer youth is required to better understand their experiences.

---

## Introduction

Newcomer populations, particularly youth, face high rates of unmet sexual and reproductive health (SRH) care needs [1,2], which includes contraception care. Unmet health needs refer to the difference between the services patients require to manage a health problem and the services that they actually received [3]. This includes the newcomer youths' subjective perception of the quality of the care received, as well as the barriers they face when accessing SRH care in Canada.

In a systematic review that synthesized the literature on 'the unmet healthcare needs among migrant populations in Canada', Chowdhury and colleagues (2021) identified sexual and reproductive health services, including contraceptive care, as one of the five areas of unmet needs for immigrants and temporary migrants in Canada [4]. A survey of immigrant youth (n = 1216) living in Toronto revealed that although immigration status on its own does not influence access to SRH services, related factors such as race, culture, income, and social location play a role in determining access to SRH services among immigrant youth [2].

Results from a 2023 qualitative study with 78 immigrant participants and 10 immigrant-focused service providers in British Columbia reported that many participants faced high unmet needs for primary care services. This included SRH services and contraception care, due to a lack of health care coverage [5]. Throughout Canada, initiatives to address barriers for newcomer youth have concentrated on eliminating the costs associated with contraception access [6]. For example, the British Columbia prescription contraception subsidy was introduced in April 2023 to address cost-related barriers to contraception [7]. This subsidy requires a provincial health card and covers the cost of many contraceptives, except non-prescription methods (such as condoms) [7]. British Columbia's 'Plan Z' also covers contraceptives for eligible newcomers to the province during the three-month wait period before provincial health coverage begins [8].

Current immigration and temporary migration trends suggest a steady and continual growth of newcomers in Canada, many newcomers being 15–24 years old when they first arrive [9]. In 2021, almost one in four people in Canada were identified as a landed

immigrant or a permanent resident, making up 23% (8.3 million) of the total population [10]. The immigrant population in Canada is projected to continue to grow and will make up 29.1% to 34% of the population by 2041 [10]. In addition, almost 6.4 million Canadians are considered second-generation immigrants, with at least one foreign-born parent, and make up 17.6% of the total population. Similar trends of growth have been documented among non-permanent residents in Canada. The 2021 census data enumerated 924,850 non-permanent residents, which includes international students, individuals on temporary work permits and asylum seekers [11]. Among non-permanent residents, 39.3% were 0–25 years old [11]. The growing number of newcomer youth in Canada underscores the importance of addressing their unmet needs for contraception care.

Barriers to newcomer youth's access to contraception care may be attributed to factors such as cultural beliefs, cost of accessing care, and systematic barriers such as insurance coverage. However, our knowledge of how these factors influence newcomer youth's access to contraception care in Canada is limited. The purpose of this scoping review is to summarize the current qualitative evidence of newcomer youth's experiences with accessing contraception care and highlight gaps in the evidence. Our aim was to inform the scope, design, and inclusion criteria for future qualitative research involving newcomers and contraception access in Canada.

## Materials and methods

### Study design

We followed the Joanna Briggs Institute (JBI) guidelines [12] to conduct a scoping review of published indexed literature and published theses. We followed the PRISMA-ScR guidelines to report the results of this scoping review (S1 Checklist PRISMA-ScR Checklist) [12]. The JBI guidelines include the application of the PRISMA-ScR. The JBI guidelines offer flexibility in the data analysis process [12]. We reported the findings of the analysis as a narrative summary of the included articles.

### Sources

To include all relevant literature on this topic, we searched five relevant databases (MEDLINE, Embase, CINAHL, PsycINFO, Web of Science) for indexed studies on June 14, 2023. We limited the grey literature to published theses to capture any recent literature that had not yet been published in scientific journals. Thesis searches were run in Theses Canada, and UBC Library archives for relevant literature. Further, we conducted citation searching by reviewing the reference lists of highly relevant studies to identify further studies.

### Search strategy

Our review team included a PhD expert in qualitative methods and access to family planning services (SM), graduate students (VP, ZK), and a clinician-scientist (BO). The lead and senior author (SM, ZK) designed and provided oversight for the scoping review and worked closely with a subject librarian (VK) to develop and refine the search strategy (S2 File Search Strategy). We reviewed the search strategy of a recently published scoping review on abortion and contraception for incarcerated people in Canada and referred to their search terms for contraception as the initial basis for the scoping review search strategy [13]. Building on this, our search terms included a combination of 'emigrants, immigrants, undocumented immigrants, refugees, transients, immigration, asylum seeker, displaced person, incomer, newcomer, migrant, and resettler' and 'adolescent, middle school, pubescent, juvenile, teen, youth, high school, young adult' to define the population of interest. We used an unvalidated search filter developed by the University of Alberta to limit our search to studies that involved residents of Canada [14].

### Inclusion criteria

Qualitative studies that described newcomer youth's experiences of contraception care in Canada were eligible for this scoping review. Studies published before 2010 were not included, as we wished to focus on the current health system

context. Available methods and attitudes toward contraception have changed over the last 13 years and we anticipated that studies published before 2010 will not provide an accurate depiction of the current experiences of youth accessing contraception care. Studies that applied qualitative methods such as focus groups, individual interviews, qualitative surveys, and arts-based methods met the inclusion criteria. We excluded conference proceedings, preprints, policy reports, and whitepapers from the grey literature search. We also excluded any study design that did not focus primarily on use of qualitative methods.

For the purposes of this scoping review, we defined newcomer youth as individuals who reside in Canada as temporary residents (for example, on a limited study permit visa), refugees, permanent residents as well and undocumented residents. Canada defines an 'immigrant' as a person who has been given the right to live permanently in Canada [15]. Both immigrants, as defined by Canada, as well as individuals who are not considered immigrants but self-identify as newcomers to Canada were eligible for participation in this research. In this review, youth refers to individuals aged 15–25 years. This upper limit of 25 is used frequently in Canada as the upper range for contraceptive subsidy programs for youth [16], contraception guidelines for youth [6,17], and surveys that analyze data on youth contraception access [18,19]. Therefore, studies that only involved children under 15 years or adults over 25 years old were excluded from this scoping review (S3 File Reasons for Excluding Studies).

The concept of 'contraception access' included intersecting factors such as knowledge of contraception, attitudes towards contraception, cultural and religious positioning, gender, transportation, insurance, and cost. Studies that explored SRH but did not mention contraception were excluded from the scoping review. Only studies conducted in English were included. See Table 1 for inclusion and exclusion criteria.

## Screening, study selection and data charting

After two co-authors (BO, ZK) conducted the searches, all identified citations were included in Covidence, a web-based software developed by the Cochrane group to streamline evidence synthesis reviews [20]. We used Covidence to manage independent title/abstract screening, full text screening, data extraction, and risk of bias assessment, after which we used Excel spreadsheets to manage data extraction, analysis, and synthesis. Two co-authors (BO, VP) independently screened the literature. They met to discuss any discrepancies in the screening process. A third reviewer (ZK) met to resolve any conflicts that arose during this process. Two co-authors (BO, VP) conducted data extraction for included studies. After

**Table 1. Inclusion and exclusion criteria for the scoping review.**

|  | Inclusion | Exclusion |
|---|---|---|
| **Population** | Newcomer youth (immigrants, refugees, temporary residents, undocumented migrants); aged 15–25 years | Studies that focus on children younger than 15 years old and individuals older than 25 years old |
| **Concept** | Access to contraception (including all forms of contraception). Access to contraception includes factors that influence access, such as knowledge of contraception, attitudes towards contraception, culture and religion, gender, transportation, insurance, and ability to pay for services related to contraception | Studies that focus on sexual and reproductive health but do not mention contraception or any form of birth control; studies that do not describe the experiences of newcomer youth; studies on newcomer youth that do not include contraception |
| **Context** | Studies that include newcomer youth (15–25 years) living in Canada | Studies conducted outside Canada |
| **Types of evidence source** | Indexed publications and theses that used qualitative research methods | Research that primarily used quantitative research methods, literature reviews, study protocols, conference proceedings, preprints, policy reports, whitepapers, clinical trials, and clinical practice guidelines |
| **Language** | Studies published in English | Studies published in languages other than English |
| **Year** | Studies published during and after 2010 | Studies published before 2010 |

which, two authors (ZK, BO) conducted qualitative analysis, synthesis, and interpretation. The senior author (SM) provided feedback and guidance throughout the research process.

We adapted relevant headings for our data charting table from the JBI guidelines and a scoping review on adolescent health [12,21]. The data charting table consisted of the following headings: study title, authors, study objectives, methods, context or setting, participant characteristics, data analysis, and authors' conclusions (S4 Data Charting Table). Data extraction of the included studies was completed by one team member (VP), while a second member (BO) reviewed the extractions, consistent with the JBI scoping review guidelines.

### Analysis and synthesis

Finally, the review team (ZK, BO, SM) conducted a descriptive qualitative synthesis of the included studies. We re-read the included studies and data charts to summarize the qualitative methods, characteristics of participants, and descriptions of youth engagement in the research process. We summarized and synthesized study results related to contraception care. To ensure that we were accurately including all relevant data related to contraception access, we thoroughly reviewed each article and took note of instances where the author or a participant's quote explicitly mentioned 'contraception', 'birth control', 'condoms,' or another method of contraception.

## Results

Our search yielded 577 studies from the indexed literature and 14 theses from our grey literature search (see Fig 1). We identified no additional studies through citation searching. After excluding duplicates, we screened 415 titles and abstracts and retrieved 17 eligible studies for full-text review. Five texts met the inclusion criteria and were included for analysis (see Table 2).

### Data collection approaches

Boafo-Arthur (2013) conducted individual interviews to explore perspectives about sexuality among Ghanaian youth living in Canada [22]. In this thesis, the author considers the physical and social settings, and parental or caregiver attitudes with an ethnographic lens and explores how these factors influence Ghanaian youth's perspectives about sexuality [22].

Ashdown et al. (2015) evaluated an SRH education intervention by conducting eight focus groups with youth [23]. The authors delivered a workshop that consisted of popular teaching strategies. This was followed by a focused group discussion to explore youths' perception of commonly used methods to teach SRH, such as games, lectures, and hands-on demonstrations [23]. The authors used an inductive analysis framework to identify themes emerging from the focus groups [23].

Taylor et al. (2022) also developed and evaluated an SRH education intervention – a theatre-based sex education workshop [24]. Immigrant youth were recruited and trained as peer educators in this interactive workshop [24]. The authors used mixed methods including surveys, focus groups, peer interviews, and scene creation (peers role-playing and creating scenes related to the topic) to elicit qualitative data that evaluated the workshop [24]. The authors used thematic analysis to generate key themes from the data [24].

In the first of Meherali et al.'s two studies (2021), authors explored South-Asian immigrant adolescents' experiences with access to SRH services in Alberta, Canada [25]. A doctoral student who led the study conducted four in-person focus groups and two virtual focus groups on the videoconferencing platform, Zoom [25]. Following the data collection, the authors applied thematic analysis to the study data [25].

Meherali et al. (2022) conducted another qualitative inquiry to better understand the SRH needs of immigrant adolescents in Canada [26]. In this study, the authors conducted twenty 45-minute individual interviews on the videoconferencing platform Google Meet, again analyzed through the lens of inductive thematic analysis [26].

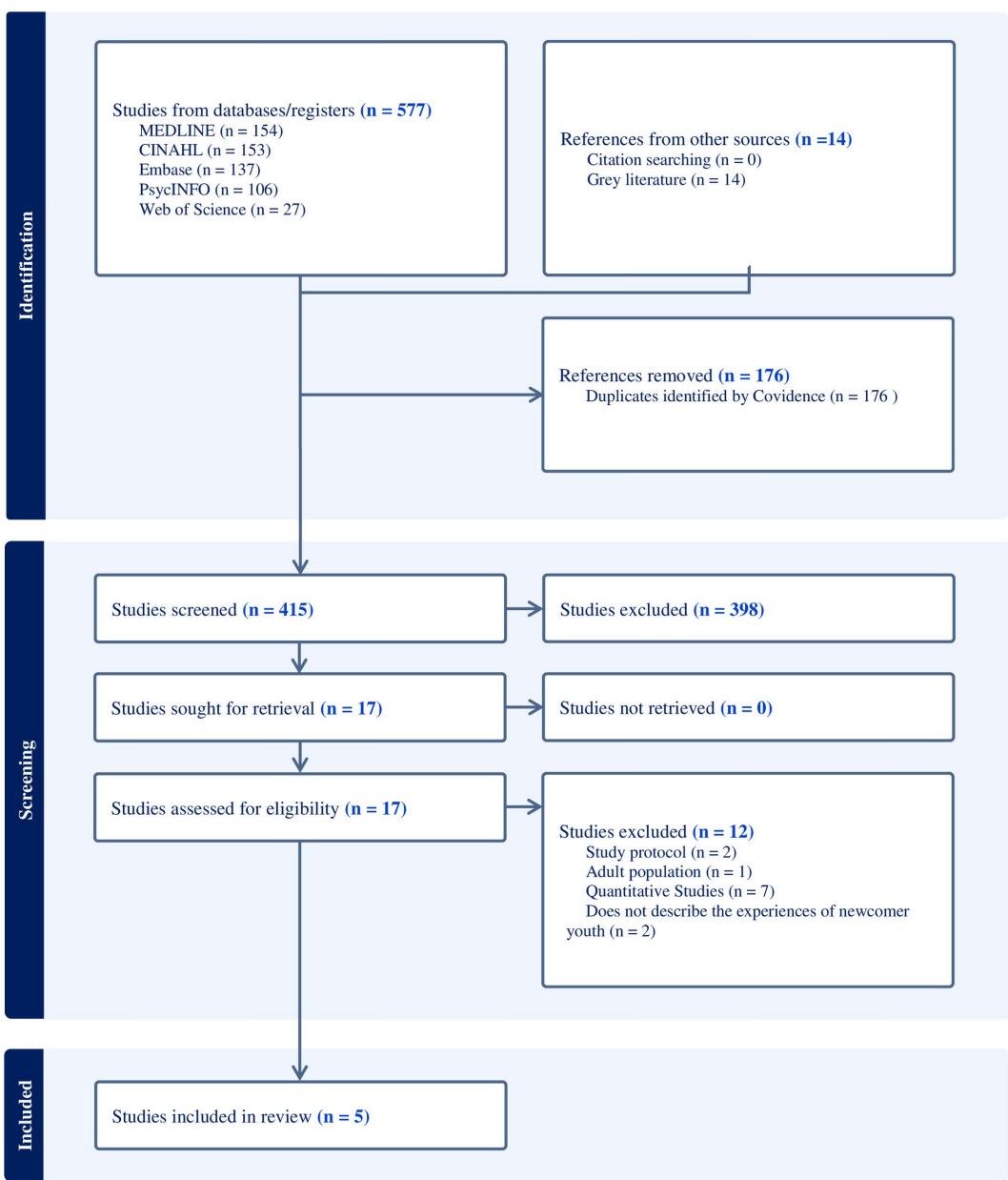

**Fig 1. Prisma diagram.**

## Participant characteristics

Boafo-Arthur (2013) interviewed 15 Ghanaian youth, who were students at Brock University (Niagara, Ontario), and had been living in Canada for a period of 1–13 years [22]. Five participants were female, and 10 participants were male. Participants were 18–27 years old [22].

The Ashdown et al. (2015) study had 80 youth participants, aged 15 and 19 years [23]. Participating youths were newcomers and street-involved. However, the authors did not collect demographic information on the participating newcomer youth [23]. The authors also did not report the total number of newcomer youth participants in the study [23]. The authors

**Table 2. List of included studies (n = 5).**

| Title | Authors | Study objectives | Methods | Participants | Key findings related to contraception |
|---|---|---|---|---|---|
| Exploring Perspectives about Sexuality among Ghanaian Youth Living in Canada: Understanding the Impact of Cultural Contact | Boafo-Arthur (2013) | *'This qualitative study examined how social contact with Canadian society has impacted the views of Ghanaian students living in Canada about sexuality.'* | This is research is part of a thesis. The author used semi-structured, individual interviews to collect data. | 15 Ghanaian youth participated in this study. Five participants were female, and ten participants were male. Participants were 18–27 years old. All participants were students at Brock University (Niagara, Ontario), and had been living in Canada for a period of 1–13 years. | Ghanaian youth expressed that they had varying degrees of knowledge of contraception. Some youth were aware of various contraceptive methods and found contraception to be generally more accessible in Canada when compared to Ghana. |
| Youth perspectives on sexual health workshops: informing future practice | Ashdown et al. (2015) | *'Newcomer and street-involved youth provided their perspective on the design and content of a sexual education workshop.'* | Focus group discussion following an educational workshop intervention. An inductive analysis framework was applied to the data. | 80 youth participants aged 15–19 years. Half of the participants were newcomer youth. Participants were from a Canadian city (the name of the city was not disclosed in the study). | Participants described the importance of language and discussed their preference for using English for words such as penis, sex as they are not widely used in the language, they speak in their birth countries. |
| Barriers to and facilitators of South Asian immigrant adolescents' access to sexual and reproductive health services in Canada: A qualitative study | Meherali et al. (2021) | *'To explore the barriers to and facilitators of South Asian immigrant adolescents' access to SRH services in Alberta.'* | Four in-person focus groups and two virtual focus groups were conducted on Zoom. Each focus group discussion lasted 60–75 minutes. The authors applied thematic analysis to the study. | 42 immigrant adolescents aged 14–20 years. 36 participants were female, and 21 participants were university students. 31 participants lived in Edmonton, Alberta with their parents. | Participants described receiving varying forms of SRH education – some of which covered contraception. Twelve participants reported feeling unsatisfied with school-based education and were unprepared to access contraception due to their lack of knowledge on the topic. |
| Sex Education by Theatre (SExT): the impact of a culturally empowering, theatre-based, peer education intervention on the sexual health self-efficacy of newcomer youth in Canada | Taylor et al., (2022) | *'This study involved the design, implementation and evaluation of a novel and innovative participatory action research project, SExT: Sex Education by Theatre.'* | Surveys, focus groups, peer interviews and arts-based data collection. Guided by Participatory Research Action principles. Thematic analysis was used to identify key themes emerging from the data. | 19 first or second-generation immigrant youth aged 15–20 years. 10 participants were female, and 9 participants were male. Participants lived in Thorncliffe Park in Toronto, Canada. | Participants expressed a lack of confidence in their ability to communicate about contraception with their partners. A culturally appropriate SRH education intervention, which provided them with the knowledge about contraception as well as the opportunity to interact with peers of a different gender improved their confidence in communicating about contraception. |
| Understanding the sexual and reproductive health needs of immigrant adolescents in Canada: a qualitative study | Meherali et al. (2022) | *'To explore the SRH information needs of immigrant adolescents in the province of Alberta, a major destination for immigrant populations in Canada.'* | Twenty 45-minute individual interviews were conducted on Google Meet. An adolescent advisory group provided feedback on the research process. The authors performed an inductive thematic analysis of the data. | 21 immigrant adolescents aged 14–19 years. Twenty of the participants were female, and all participants were first-generation immigrants. The participants lived in Alberta. | Participants discussed how the parental consent required to see health care providers is a barrier to contraception access. The authors described the sociocultural context of immigrant families that intersects with contraception access. Participants also shared about learning about contraception from peers. |

defined newcomer youth as '*young people who are recent immigrants and/or refugees to Canada*', however, they did not provide a definition of 'street-involved' youth [23]. The youth who participated in this study lived in a Canadian city. The name of the city was not disclosed in the study [23].

The study Taylor et al., (2022) conducted included 19 participants, with 10 participants who were female-identified and 9 who were male-identified [24]. Eighteen participants identified as a person of colour and over two-thirds of the participants were first-generation immigrants [24].

The focus groups conducted by Meherali et al. (2021) to explore the barriers and facilitators of SRH information among South-Asian immigrant adolescents, involved 42 immigrant adolescents aged 14–20 years [25]. Among them, 36 participants were female, and 21 participants were university students. Thirty-one participants lived in Edmonton, Alberta, with their parents [25].

Twenty-one adolescent immigrant youth participated in the study conducted by Meherali et al. (2022) [26]. The ages of the participants ranged from 14 to 19, with the majority (n = 18) participants being 18–19 years old. Only one of the participants was male [26]. The authors provided a detailed breakdown of the demographic characteristics of participants, including their birth countries and the languages they spoke [26]. Six participants were from India, two participants were from the Philippines and two participants were from Bangladesh [26]. One immigrant adolescent was from each of these countries China, Bahrain, Nigeria, Nepal, Venezuela, Colombia, Saudi Arabia, Malaysia, Kuwait, Ireland, and Kazakhstan [26]. Thirteen participants lived in Canada for over ten years at the time of participation [26].

## Youth involvement in research

Two of the five studies included in this review created opportunities for meaningful youth involvement in the research process [24,26].

Ashdown et al. (2015) did not report the involvement of youth in the research process [23]. However, the authors described the involvement of youth-serving organizations, which helped the researchers reach immigrants and refugees from low-income countries [23]. Similarly, the studies conducted by Boafo-Arthur (2013) and Meherali et al. (2021) also did not report newcomer youth engagement in the research process [22,25].

Taylor et al., (2022) utilized a participatory action research approach, characterized by the explicit goal of empowering participants through the research as co-creators of the knowledge [24]. The authors took steps to mitigate barriers to youth participation in the research process and met with them regularly to ensure that youth remained active research partners [24]. In addition, the authors consulted youth partners to ensure that the project met the needs of the community where it took place [24].

In Meherali et al. (2022), the qualitative study conducted with youth participants aged 14–19 involved an adolescent advisory group consisting of immigrant adolescents, who reviewed and provided feedback on the interview guides [26]. One member from the adolescent advisory group also participated in seven of the 21 interviews that were conducted for this study, to observe and provide feedback on the interview process [26]. The adolescent advisory group met once a month with members of the research team, where they shared their insights on the research process [26].

## Contraception access

Boafo-Arthur (2013) reported that Ghanaian youth living in Canada varied in their awareness and knowledge of contraception care [22]. One participant shared that although Ghanaian youth are aware of their contraception options, Ghanaian cultural influences intersect with their ability to obtain contraceptives. This participant [22] shared,

*'I think they [youth] are aware. I was aware as early as when I was in my early teens. I knew of condoms, though it's not that I was using them but I knew of condoms and all those things. But, I think that people just don't seem to like using them, because even if you are a youth and you go into a pharmacy to go and buy condoms, or even to a mall*

*to go and buy condoms people are going to look at you in some weird way. If you meet an elderly person, you will be shocked that the elderly person might not even sell it to you.'*

Participants in this study agreed that compared to Ghana, it is easier for them to access and learn about contraception in Canada [22]. One participant's experience illustrates that the ease of contraception access in Canada is related to less stigma about sexual activity [22]:

*'A Ghanaian will go to the pharmacy shop to buy condoms and get there and there are people there minding their own business and you can't even tell the person you are buying the condom. Why? Because they think you're gonna have sex. When I came here the first month, I went to a convenience shop to buy something, and this guy just came in a loud voice, "I want the blue Trojan", in front of everybody. Like he said it so loudly, "I am buying a condom". You can never see anyone doing that in Ghana. That's one huge difference. The awareness is there, but the perception of someone going to buy a condom, you are already condemned like you are a "whore", you are a "player", you like sex, you are "spoilt".'*

This attitude was reflected by another participant, who shared they had limited knowledge of contraception methods when they arrived in Canada, but have had opportunity to learn more about options since their relocation [22]. To that end, another participant shared,

*'In Ghana everyone just thinks of condom, condom use. It wasn't until I got here that I learnt really about contraception and the after-morning pill that they do here. So I don't think that there is enough awareness.'*

In the evaluation of an SRH education workshop conducted by Ashdown et al. (2015), newcomer youth shared that language was an important factor in SRH education, and suggested ways to make it easier for participants who speak English as a second language. Their suggestions included practical steps, such as speaking slowly and allowing time for participants to comprehend the educational content. Youth participants further elaborated that the language used to communicate topics related to SRH had cultural implications beyond comprehension [23]. For example, some newcomer participants shared that words like penis, oral, and sex are not widely used in their first language, and expressed feeling more comfortable discussing these terms and related topics in English [23].

Most of the newcomer participants agreed that while SRH workshop facilitators do not have to be of the same cultural background as them, they should be familiar with the cultural backgrounds of the workshop participants [23]. In this study, Ashdown et al. (2015) also reported that newcomer participants found it challenging to reconcile what they had learned about SRH and contraception in their home countries, and what they learned in Canada [23], particularly involving abstinence education:

*'ABC: Abstinence, be faithful and the last, if you must, is use condoms. And here [in Canada], it changes so, condoms first and then the others.'*

Continuing with a focus on education, Meherali et al.'s (2021) focus group investigation of the SRH information and service access barriers faced by South Asian immigrant adolescents aged 14–20 years identified that immigrant adolescents had mixed experiences with SRH education in school [25]. Contraception care was discussed in the context of school-based education, where eighteen participants shared that they received SRH education in school and learned about contraception care through school-based SRH education in Canada or their country of birth [25]. One participant expressed,

*'As SA [South-Asian] girl I am not comfortable talking with my mom about SRH... so school classes really helped me to understand male and female reproductive anatomy, fertilization, pregnancy, contraception etc.'*

In contrast, twelve participants shared that they were not satisfied with the education that they received in school. One newcomer youth shared that they did not receive information about how to access SRH services [25],

> 'In sexual health education classes, we mainly talked about periods, wet dreams, and briefly about pregnancy, STIs etc. I was never told about where to get tested for an STI, from where we can get emergency contraception, or learned where to get free condoms or contraception or where to access birth control…nothing was talked about in-depth.'

In Taylor et al.'s evaluation of a theatre-based sex education workshop (2022), the authors posit that newcomer youth may lack confidence and knowledge of contraception, hindering their ability to communicate about contraception with their partners [24]. The authors described this theme as 'protection self-efficacy', which refers to '*one's ability to effectively access, communicate about and ultimately use contraception with sexual partners*' [24]. According to the authors, most participants identified 'safer sex' to be one of the key learning from their participation in the workshop intervention [24]. In this study, newcomer youth participants also attested that interactions with peers of a different gender helped build their confidence in communicating about contraception [24]. One newcomer participant shared that their confidence in communicating with her partner about contraception improved following her participation in the workshop [24]:

> 'I was a really shy person, like before the programme, right? But now I feel I'm okay with talking to guys about sex... I don't think I would be shy... I'll tell a guy things like when to use a condom. I'd be like, 'Bro, no, you should use a condom. Like you have to.' See? We're learning.'

The same participant further elaborated that she intended to discuss their use of oral contraceptive pills (birth control pills) with her partner,

> 'I have seen changes in myself, and I was actually planning to get on the birth control pill... My boyfriend and I have talked about it, and we think it's a good idea.'

Although contraception made up a small section of the SRH findings reported in this study, it highlights an important aspect of the newcomer youth's ability to communicate with partners, which can improve through new knowledge about contraception and SRH [24].

In the interview study conducted by Meherali et al. (2022), the authors described the structural barriers that immigrant adolescents aged 14–19 years old and living in Alberta, Canada faced when accessing SRH care [26]. This study focussed on the challenges that immigrant adolescents faced when accessing all forms of SRH services, including contraception care [26]. The authors draw connections between access barriers and the sociocultural context of immigrant youth families. Specifically, they highlight how parents' attitudes toward SRH services influence newcomer youths' abilities to access birth control [26].

This study also brings attention to the role of peers as a source of information about contraception [26]. Immigrant adolescent participants cited friends and peers as the second most popular source of information for SRH after the internet and media [26]. For example, one newcomer participant shared that,

> 'I had a couple of friends that were on birth control, so I asked them about it, and they gave me the pros and cons of it, so that's kind of what helped make my decision. I'd probably say my friends.'

## Discussion

We conducted a scoping review of the qualitative literature and identified five studies on the experiences of newcomer youth in accessing contraception in Canada. Our results highlight the scarcity of published literature on this topic. Although

access to contraception care was considered in the five included studies, in none was it the primary focus of the research. The results of our scoping review provide preliminary insight into the contraception health service access experiences of newcomer youth. Our synthesis of the existing data underscores the role of language and sociocultural influences in shaping newcomer youth's perspectives and access. Our results provide additional insight into the role of formal and informal peer-based education as an acceptable approach for newcomer youth in gaining contraception knowledge. Two studies included in the review identified that appropriate educational interventions have the potential to empower youth to confidently navigate contraception care in Canada [23,24].

Current literature on healthcare access for immigrant populations identifies language as a critical barrier for newcomers who do not speak English as their first language [27]. The results of our scoping review provide additional insights into the language preferences of youth specifically regarding SRH topics. Newcomer youth who spoke English as a second language expressed a preference for discussing SRH topics in English. This finding is unique to newcomer youth's experiences and highlights the differences between the preferences of youth in the context of contraception discourse and the broader immigrant population's health access needs. For instance, past studies have identified that adult immigrants' language needs include access to interpreters and health care resources in the language spoken in their birth country [28]. Our results support the concept that newcomer youth may have unique needs and preferences when it comes to contraception access, such as discussing SRH-related topics in English rather than the language spoken in their birth countries [23].

The existing evidence illustrates that newcomer youth experiences of contraception care are linked to their unique sociocultural contexts. The thesis by Boafo-Arthur (2013) involving Ghanaian youth in Canada, for instance, illustrates the varying levels of societal acceptance of contraception and discourse around contraception that newcomer youth may encounter [22]. One newcomer youth shared that there was more openness and acceptance of contraception purchase in Canada compared to what they had observed in their country of birth in Ghana [22]. These findings are relevant for policymakers, educators, and community-based organizations involved in providing settlement services to newcomer families.

The results of this scoping review offer valuable insights into the specific needs of newcomer youth and highlight key factors that should be considered when designing educational programs and interventions that aim to enhance newcomer youth's access to contraception care. For instance, the SRH education workshop conducted by Ashdown et al. (2015) highlighted the importance of the workshop facilitator's familiarity with newcomer youth's cultural backgrounds in facilitating the SRH education [23]. Taylor et al.'s (2022) evaluation of a theatre-based sex education workshop demonstrated the role of peer-based education in bolstering newcomer youth's confidence in discussing SRH-related issues [24].

The present scoping review provides a unique contribution to the literature on the particular SRH needs of newcomer youth in Canada. Prior research has investigated the SRH needs of immigrant and newcomer peoples of all ages in Canada. In a systematic review of the unmet needs of immigrant populations in Canada, Chowdhury et al. (2021) identified that immigrants from regions where discussing or accessing SRH is deemed a culturally and socially taboo topic experienced unmet SRH needs [4]. This assertion is supported by the findings of our current scoping review. Chowdhury et al. (2021) further suggest that access to information about contraception is lacking, and misconceptions about side effects are pervasive among immigrant and refugee populations in Canada [4]. Our scoping review points towards newcomer youth's informational needs, and linguistic preferences, as well as opportunities to leverage peer-based education on contraception.

This scoping review highlighted three key gaps in current evidence.

## Lack of focus on contraception access experiences

It is important to note that the qualitative studies included in this scoping review focused on broader SRH experiences, rather than contraception care. While there may be similarities between newcomer youth's experiences of accessing SRH services and contraception care, comprehensive evidence that highlights the nuances of contraception access for

newcomer youth remains scarce. The results of this review identified a knowledge gap in our understanding of contraception access for newcomer youth. Future studies should focus on examining the needs of newcomer youth pertaining to contraception specifically.

### Lack of involvement of newcomer youth in the research process

Only two out of five studies included in this review describe youth engagement in the research process [24,26]. Taylor et al.'s (2022) participatory research action study design allowed youth to become 'co-creators' of the research evidence [24]. In Meherali et al. (2022), the authors invited immigrant adolescents to provide feedback on the interview guides and the interview process by creating an adolescent advisory group for the research. There is strong evidence to suggest that involving youth as research partners leads to the production of richer data and greater access to youth participants [29]. For instance, Wilkinson and colleagues demonstrated the success of engaging with youth aged 15–17 years in the co-development of a contraception navigator program for youth living in Indiana, USA [30]. Similarly, another qualitative study which involved Zambian youth (15–24 years) in a participatory research process to investigate their SRH needs provided further evidence to support the importance of youth engagement in the research [31]. Given this evidence, future studies that aim to explore newcomer contraception experiences should consider engaging newcomer youth in the research process to elicit rich data.

### Further exploration of the role of peers in contraception access

In their respective studies, Taylor et al. (2022) and Meherali et al. (2022) demonstrate the role of peers in empowering newcomer youth to engage in discussions about contraception and sharing information about contraceptive methods [24,26]. This is consistent with global literature on the impact of peer-led SRH education for youth [32,33]. However, considering that the nature of peer relationships may vary between first-generation newcomer youth and Canadian-born youth, further investigation of the role of peers in influencing contraception choices is warranted. For instance, newcomer youth may encounter contradictory perspectives and discourse on contraception between peers in their birth country and Canada, as highlighted in studies included here [22,23]. As a result, it is crucial to understand *where* and *how* peers influence newcomer youth's attitudes and awareness of contraception. This knowledge is essential to the development of solutions that are effective in supporting youth's abilities to access contraception care.

One of the limitations of this review is that it does not incorporate all potential grey literature on this topic. The inclusion of grey literature can provide additional context, and practical insights from diverse perspectives such as government, and non-profits and capture ongoing studies on this topic. The search strategy designed to retrieve published studies from the databases could be further strengthened by including additional keywords to include all terms for different methods of contraception available in Canada. We did not conduct a critical appraisal of the studies included in this review. Although critical appraisals are not required for scoping reviews, including a critical appraisal of the studies could potentially improve the rigour and reliability of the data. Our scoping review excluded studies published before 2010. As a result, our scoping review does not capture *how* newcomer youth's attitudes toward contraception have changed over the decade.

This scoping review describes the current evidence on newcomer youth's experiences of accessing contraception and identifies key gaps in research. Lack of research that focuses on contraception experiences specifically, and limited involvement of youth in the research process, restrict our understanding of newcomer youth's experiences of contraception access. Further research is needed to gain a deeper understanding of how cultural, religious, and parental attitudes toward contraception care intersect and influence newcomer youth's access. Conducting research that focuses on youth contraception experiences and involves youth in the research process can generate further evidence and support the nuanced understanding of these issues.

In this scoping review, we have highlighted the role of peers in making contraception care accessible to newcomer youth [24,26]. There is an opportunity to further explore the role of peers beyond making youth aware of a contraception method, which could provide essential insights into the factors that allow youth to make decisions about contraception. Similarly, examining the various factors that determine 'access', such as the interplay between cost, insurance policies, transportation, geographical location of the participants, social and environmental factors and newcomer youth's ability to access contraception care can provide a holistic understanding of how newcomer youth navigate contraception care [34]. Addressing these research gaps can help policymakers, educators, and healthcare providers develop programs, design policies and generate knowledge translation tools that are grounded in newcomer youth's needs.

Canada is home to over 8.3 million landed immigrants [10]. As Canada welcomes more immigrants, this percentage is expected to increase and represent 29% to 34% of the Canadian population by 2041. According to Statistics Canada, youth make up 10.9% of the recent immigrant population in Canada [10]. Given that Canada is experiencing a steady growth in the immigrant youth population, this study highlights that it is pertinent that contraception care, which is an essential component of reproductive autonomy, is delivered in a manner that is acceptable and accessible by newcomer youth.

## Conclusion

We conducted a scoping review of the qualitative literature to describe the experiences of newcomer youth accessing contraception in Canada. Our findings indicate that newcomer youth's contraception access needs are characterized by their unique language preferences, socio-cultural norms, and influences from peers. Future studies that focus on contraception needs, explore peer influences and involve newcomer youth as research partners can further elicit a nuanced understanding of how these factors intersect with newcomer youth's contraception access.

## Supporting information

**S1 Checklist. PRISMA-ScR Checklist.**
(DOCX)

**S2 File. Search strategy.**
(DOCX)

**S3 File. Reasons for excluding studies.**
(DOCX)

**S4 Data. Charting table.**
(XLSX)

## Acknowledgments

We acknowledge the contributions of Dr. Wendy V. Norman and Dr. Skye Barbic, who reviewed and provided feedback to a version of this scoping review that was published as a chapter in a Master's thesis. In addition, we acknowledge the contributions of Vanessa Kitchin, Knowledge Synthesis & Medical Librarian at the University of British Columbia, who guided the development of the initial search strategy.

## Author contributions

**Conceptualization:** Zeba Fariha Khan.

**Data curation:** Bimbola Olure, Victoria Paller.

**Formal analysis:** Bimbola Olure, Victoria Paller.

**Writing – original draft:** Zeba Fariha Khan.

**Writing – review & editing:** Zeba Fariha Khan, Bimbola Olure, Victoria Paller, Sarah Munro.

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
