## [Decision Letter · Decision Letter 0]

7 May 2024

PONE-D-24-05128

Newcomer youth’s access to contraception care in Canada: A scoping review of qualitative evidence

PLOS ONE

Dear Dr. Munro,

Thank you for submitting your manuscript to PLOS ONE. After careful consideration, we have decided that your manuscript does not meet our criteria for publication and must therefore be rejected.

Specifically:

The primary concern with this paper is the rationale behind exclusively incorporating qualitative studies for a scoping review. While a scoping review aims to map out and summarize existing literature on a specific topic, offering an overview of available evidence, it would be more comprehensive if quantitative studies were also included. Examining access to contraception using various study designs would allow for a more thorough exploration of research gaps and better inform policy and practice. Additionally, it is noteworthy that the authors only utilized four databases and limited the search for grey literature to published theses. Conducting a more comprehensive search is advisable for a scoping review.

I am sorry that we cannot be more positive on this occasion, but hope that you appreciate the reasons for this decision.

Kind regards,

Rogie Royce Carandang, RPh, MPH, MSc, PhD

Academic Editor

PLOS ONE

Additional Editor Comments:

Minor comments:

1. For clarity, the abstract should explicitly mention the results of the five studies in the scoping review.

2. In the introduction, it is essential to present the latest data when discussing demographic trends in Canada.

3. The rationale for conducting this systematic review needs clarification. Authors should justify why a scoping review is chosen and why qualitative studies are exclusively focused on.

4. The discussion highlighted the scarcity of published literature on the topic. Therefore, the authors should have included multiple study designs, as this aligns with the purpose of a scoping review rather than focusing solely on one study design.

Reviewers' comments:

Reviewer's Responses to Questions

**Comments to the Author**

1. Is the manuscript technically sound, and do the data support the conclusions?

Reviewer #1: Yes

Reviewer #2: Yes

2. Has the statistical analysis been performed appropriately and rigorously?

Reviewer #1: Yes

Reviewer #2: N/A

3. Have the authors made all data underlying the findings in their manuscript fully available?

Reviewer #1: Yes

Reviewer #2: Yes

4. Is the manuscript presented in an intelligible fashion and written in standard English?

Reviewer #1: Yes

Reviewer #2: Yes

Reviewer #1: The following are the minor revisions that you may want to review:

27 Embase, CINAHL, PsycINFO). Of 415 identified [insert 'studies'] through the search process, five [delete

'studies'] were

28 included in this scoping review following screening for eligibility. We [replace 'report' with 'reported'] the findings of

our

33 seeking general health care. We also [replace 'provide' with 'provided'] insights on formal and peer-based

educational

34 interventions acceptable to newcomer youth. [replace 'Literature' with 'Literatures'] that qualitatively

[replace 'describes' with 'describe'] newcomer

35 youth's experiences with contraception [replace 'is' with 'are'] scarce, and existing [replace 'literature' with

'literatures'] only[replace ' mentions' with 'mentioned']

39 youth [insert comma (,)] is required to better understand their experiences.

51 [delete 'different'] migrant populations in Canada’, Chowdhury and colleagues (2021) identified sexual

60 for primary care services [replace comma (,) with period (.)] [replace 'which' with 'This'] included SRH services and

contraception care [delete comma (,)] due to a lack of

73 population in Canada is projected to continue to grow and [insert 'will'] make up [replace '29%' with '29.1%' as

cited in the study] to 34% of the

81 Barriers to newcomer [insert 'youth’s'] access to contraception care may be attributed to factors such as cultural

82 beliefs, cost of accessing care, and systematic barriers such as insurance coverage, [delete 'which']

90 We followed the Joanna Briggs Institute [delete '[12]' and insert '(JBI)'] guidelines [insert '[12]'] to conduct a

scoping review of

93 PRISMA-ScR. The JBI guidelines offer flexibility in the data analysis process [12]. We [replace 'report' with

'reported'] the

99 to published theses to capture any recent [replace 'literature with 'literatures'] that [replace 'had' with 'has'] not

yet been published in scientific

122 published before 2010 [replace 'would' with 'will'] not provide an accurate depiction of the current experiences of

155 included studies [replace comma (,) with period (.)] [Capitalize the word 'after'] which [insert comma (,) and 'two

authors (ZK, BO) conducted'] qualitative analysis, synthesis, and interpretation [insert period (.) and delete 'were

conducted']

156 [delete '(ZK, BO).'] The senior author (SM) provided feedback and guidance throughout the research

163 was completed by one team member (VP), while a second [insert 'member'] (BO) reviewed the extractions,

199 strategies [replace comma (,) with period (.) and replace 'which' with 'This'] was followed by a [replace 'focus'

with 'focused'] group discussion to explore youths’ [replace 'perceptions' with 'perception'] of

206 educators [delete 'who participated'] in this interactive workshop [24]. The authors used mixed methods

214 videoconferencing video conferencing platform [insert comma (,)] Zoom [25]. Following the data collection, the

authors applied

219 minute individual interviews on the videoconferencing platform [insert comma (,)] Google Meet [replace comma

(,) with period (.) and insert 'The authors'] again analyzed

229 Participating youths were newcomers and street-involved [replace comma (,) with period (.) and capitalize

'however'], the authors did not collect

234 participated in this study lived in a Canadian city[replace semicolon (;) with period (.) and capitalize 'the'] name of

the city was not disclosed in the

237 [replace 'The ‘Sex Education by Theatre (SExT)' with 'The study Taylor et al., (2022) conducted'] included 19

participants, with 10 participants who

253 Venezuela, [replace 'Columbia' with 'Colombia'], Saudi Arabia, Malaysia, Kuwait, Ireland, and Kazakhstan [26].

Thirteen

263 newcomer youth engagement in the research process [insert references '[22, 25]'.

285 'I think they [youth] are aware [replace semicolon (;) with period (.)] I was aware as early as when I was in my

early

287 condoms and all those things [replace semicolon (;) with period (.) and capitalize 'but' then insert comma (,)] I

think that people just don’t seem to like

288 using them [replace semicolon (;) with period (.) and capitalize 'because'] even if you are a youth and you go into

a pharmacy to go

289 and buy condoms [insert comma (,)] or even to a mall to go and buy condoms people are going

290 to look at you in some weird way [replace semicolon (;) with period (.) and capitalize 'if'] you meet an elderly

person, you will be

299 person you are buying the condom [replace semicolon (;) with period (.) and capitalize 'why']? Because they think

[replace 'you' with 'you’re'] gonna have

300 sex [replace semicolon (;) with period (.) and capitalize 'when'] I came here the first month, I went to a

convenience shop to buy

301 something, and this guy just came in a loud voice, [Quote "I want the blue Trojan"], in

302 front of everybody [replace semicolon (;) with period (.) and capitalize 'like'] he said it so loudly, [Quote “I am

buying a condom”, replace semicolon (;) with period (.) and capitalize 'you'] can

303 never see anyone doing that in Ghana. [capitalize 'that’s'] one huge difference [replace semicolon (;) with period

(.) and capitalize 'the']

304 awareness is there [insert comma (,)] but the perception of someone going to buy a condom, you

311 'In Ghana everyone just thinks of condom, condom use, [delete duplicate 'condom use', replace semicolon (;) with

period (.), capitalize 'it' , delete 'was' and replace with 'wasn’t']

312 [delete 'not'] until I got here that I learnt really about contraception and the after-

313 morning pill that they do here [replace semicolon (;) with period (.) and capitalize 'so'] I don’t think that there is

enough

326 Most of the newcomer participants agreed that [replace 'of' with 'while'] SRH workshop facilitators do not have to

be

327 of the same cultural background as them, [delete 'however,'] they should be familiar with the cultural

441 backgrounds in facilitating the SRH education [23]. Taylor et al.’s [insert year of publication '(2022)'] evaluation of

a theatre-based

447 needs of immigrant populations in Canada, Chowdhury et al. [insert year of publication '(2021)'] identified that

immigrants from

450 scoping review. Chowdhury et al. [insert year of publication '(2021)'] further suggest that access to information

about

509 This scoping review describes the current evidence on newcomer [replace 'youth' with 'youth’s'] experiences of

514 parental attitudes toward contraception care intersect and influence newcomer [replace 'youth' with 'youth’s']

access.

527 these research gaps can help policymakers, educators, and [Kindly spell out 'HCPs' as this was not previously

mentioned] develop programs, design

577 Paediatr Child Health. 2019;24: [correct page '160–164' to '160-169']. doi:10.1093/pch/pxz033

Reviewer #2: Thank you for the opportunity to review the manuscript by Khan et al.

This scoping review on newcomer youth's experiences accessing contraception care in Canada is an important topic. To strengthen the manuscript for publication consideration, I'd like to offer some suggestions for addressing some key areas.

1. As observed in the abstract, kindly identify the eligibility criteria used and the data extraction process. Also, please change and improve the formatting of the abstract, it should include the following sections: Objectives >> Design >> Data Sources >> Eligibility Criteria >> Data extraction and synthesis >> Results >> Conclusions. For your reference, please visit the link provided http://bmjopen.bmj.com/content/8/3/e019438

2. Keywords should be arranged in alphabetical order

3. I noticed the search strategy relied solely on keywords. It's generally recommended to combine keywords with MeSH terms (tw) for more focused results. To improve transparency, could you include the search strategy used for each database in the supplementary files? This would be helpful, including the initial search terms and the final results after removing duplicates.I noticed the search strategy relied solely on keywords. It's generally recommended to combine keywords with MeSH terms (tw) for more focused results. To improve transparency, could you include the search strategy used for each database in the supplementary files? This would be helpful, including the initial search terms and the final results after removing duplicates.

4. To enhance transparency in line with the PRISMA 2020 Checklist, could you consider including a table in the supplementary files that details the 12 excluded studies after full-text screening? The table could list each study and the corresponding reason for exclusion. This would provide valuable insight into the selection process.

5. What is the tool you used in your data extraction, analysis and synthesis? Could you please present and discuss the specific tool and present the data in your paper? Using certain tools for each selected study design can ensure the quality of evidence and reduce the risk of bias in the conduct of your study.

6. The manuscript mentions a data charting table with specific headings outlined on lines 160-162, including study title, authors, objectives, methods, context, participant characteristics, data analysis, and authors' conclusions. However, the paper currently only presents a table listing the included studies. For improved transparency and to allow readers to fully grasp the data analysis process, including the data charting table or a similar table with the mentioned headings in the supplementary files would be beneficial. This would provide a more comprehensive picture of the data extraction and analysis.

7. To improve the flow of information, consider combining the "Data" heading and its context onto the same page as the actual data content. This will prevent unnecessary page breaks and enhance readability. For optimal formatting, it's recommended to follow the specific guidelines outlined in the PLOS One criteria for the "Data" section. This will ensure a consistent and professional presentation of your data.

**Do you want your identity to be public for this peer review?** For information about this choice, including consent withdrawal, please see our Privacy Policy

Reviewer #1: No

Reviewer #2: No

- - - - -

---

## [Author Response · Author response to Decision Letter 1]

30 May 2024

Appeal: PLOS ONE Decision: PONE-D-24-05128

Response to Editor and Reviewer's Comments

* This response to reviewers letter is also appended as a PDF file.

Thank you for providing us with the opportunity to respond to the comments that we received from the editor and the peer reviewers. We have addressed the concerns raised by the editor, as well as provided responses to the minor revisions suggested by peer reviewers 1 and 2.

We received the following comment from the editor:

'The primary concern with this paper is the rationale behind exclusively incorporating qualitative studies for a scoping review. While a scoping review aims to map out and summarize existing literature on a specific topic, offering an overview of available evidence, it would be more comprehensive if quantitative studies were also included. Examining access to contraception using various study designs would allow for a more thorough exploration of research gaps and better inform policy and practice. Additionally, it is noteworthy that the authors only utilized four databases and limited the search for grey literature to published theses. Conducting a more comprehensive search is advisable for a scoping review.'

We have provided a detailed rationale for conducting a scoping review of qualitative studies in our response to #3 and #4 of the minor comments (see below).

In addition, we wish to highlight that we searched the five indexed databases (Medline, Embase, CINAHL, PsycINFO and Web of Science). In addition, we also searched Theses Canada and UBC Library archives for relevant literature for published theses on this topic. We consulted a Knowledge Synthesis & Medical Librarian at the University of British Columbia (Vanessa Kitchin, MI) to identify relevant databases for this review, create the keyword search strategy, and run our scoping review searches.

We list the five databases in the PRISMA diagram that we submitted with our original manuscript. We had incorrectly classified this as four databases in the manuscript. We have amended this detail in the description of the data sources in the 'Abstract' and under the heading 'Sources'. These changes correspond to lines and lines 29 to 30, and lines 130 to 135 in the updated manuscript. The changes we made to the 'Sources' are also highlighted below.

'To include all relevant literature on this topic, we searched four five relevant databases (MEDLINE, Embase, CINAHL, PsycINFO, Web of Science) for indexed studies on June 14, 2023. We limited the grey literature to published theses to capture any recent literature that had not yet been published in scientific journals. Thesis searches were run in Web of Science, Theses Canada, and UBC Library archives for relevant literature.'

Below we provide a point-by-point response to the minor comments made by the editor:

1. For clarity, the abstract should explicitly mention the results of the five studies in the scoping review.

We revised the reporting of our results in the abstract to explicitly describe the findings of the five studies included in this review.

‘Results: We screened 415 titles and abstracts and retrieved 17 eligible studies for full-text review, of which five were included for analysis. Results highlight that sociocultural influences play an important role in newcomer youth's perspectives and access to contraception care. The language preferences of newcomer youth in the context of contraception are unique and differ from the needs of the broader immigrant population seeking general health care. Results also identify formal and peer-based educational SRH interventions acceptable to newcomer youth.’

2. In the introduction, it is essential to present the latest data when discussing demographic trends in Canada.

The demographic data presented in our manuscript are from the 2021 Census conducted by Statistics Canada. As the census is conducted every five years, this represents the latest demographic data available. We have reviewed recent reports from Statistics Canada for relevant information. Although there have been newer reports, they continue to utilize data from the 2021 Census. Additionally, we have provided more recent demographic data about international students, as highlighted in red:

'Similar trends of growth have been documented among non-permanent residents in Canada. The 2021 census data enumerated 924,850 non-permanent residents, which includes international students, individuals on temporary work permits and asylum seekers [11]. Among non-permanent residents, 39.3% were 0 to 25 years old [11]. The growing number of newcomer youth in Canada underscores the importance of addressing their unmet needs for contraception care.'

3. The rationale for conducting this systematic review needs clarification. Authors should justify why a scoping review is chosen and why qualitative studies are exclusively focused on.

The objective of this systematic review was to inform the scope, design, and inclusion criteria for future qualitative research involving newcomers and contraception access in Canada. We were specifically interested at mapping gaps in the existing literature for this defined research area. We selected a scoping review as it aligned best with our objective. A scoping review is a “form of knowledge synthesis that addresses an exploratory research question aimed at mapping key concepts, types of evidence, and gaps in research related to a defined area or field by systematically searching, selecting, and synthesizing existing knowledge.” We have specified this at the end of the Introduction and in the beginning of the Methods, under Study Design.

‘The purpose of this scoping review is to summarize the current qualitative evidence of newcomer youth's experiences with accessing contraception care and highlight gaps in the evidence. Our aim was to inform the scope, design, and inclusion criteria for future qualitative research involving newcomers and contraception access in Canada.’

‘We followed the Joanna Briggs Institute (JBI) guidelines [12] to conduct a scoping review of published indexed literature and published theses. We selected a scoping review as it aligned best with our objective to map the knowledge and knowledge gaps in the existing qualitative research involving newcomers and contraception access in Canada.’

Regarding the rationale for our focus on the qualitative literature, youth experiences of accessing sexual and reproductive health (SRH) – including their attitudes, beliefs, preferences, and needs – can be nuanced, detailed, and difficult to quantify. In this regard, qualitative studies are best suited to investigate what youth are thinking, and how they are feeling when accessing SRH, and why they access or do not access care depending on context. Considering the objective and scope of our research, we focused on studies that used various qualitative data collection methods (focus groups, individual interviews, qualitative surveys, and arts-based methods).

4. The discussion highlighted the scarcity of published literature on the topic. Therefore, the authors should have included multiple study designs, as this aligns with the purpose of a scoping review rather than focusing solely on one study design.

Following the Joanna Briggs Institute guidelines, “a scoping review can include any and all types of literature (eg, primary research studies, systematic reviews, meta-analyses, letters, guidelines, websites, blogs). However, reviewers may wish to impose limits based on the knowledge that particular types of sources would be most useful and appropriate.”

We followed this guidance to impose limits on knowledge sources that would be most useful and appropriate for addressing our research objective. Specifically, we limited our scoping review to include the most appropriate research studies, i.e., qualitative research, to address our objective, which was to summarize the evidence on newcomer youth's experiences with accessing contraception care to inform the scope, design, and inclusion criteria for future qualitative research involving newcomers and contraception access in Canada. The scarcity of publications on this topic area is an important finding of our review, which was determined after the completion of the scoping review. The paucity of qualitative evidence on this topic does not warrant the inclusion of quantitative studies or other methodologies that do not meet our research objective.

We wish to highlight that we selected PLOS One for submission because the journal has published qualitative reviews, including a qualitative scoping review in 2023. The study was a scoping review of qualitative studies investigating reproductive health knowledge, attitudes, and practices in a single jurisdiction, Rwanda. We reviewed this publication and considered the peer review comments. Given the publication of this article in 2023, we felt that the scope and aims of our manuscript would be an excellent fit for PLOS ONE.

Response to Reviewer 2's comments

1. As observed in the abstract, kindly identify the eligibility criteria used and the data extraction process. Also, please change and improve the formatting of the abstract, it should include the following sections: Objectives >> Design >> Data Sources >> Eligibility Criteria >> Data extraction and synthesis >> Results >> Conclusions. For your reference, please visit the link provided http://bmjopen.bmj.com/content/8/3/e019438

We have reformatted the abstract in the updated manuscript to include highlight these sections. We tracked the changes that we made to the abstract.

2. Keywords should be arranged in alphabetical order

The keywords have been re-arranged in alphabetical order.

3. I noticed the search strategy relied solely on keywords. It's generally recommended to combine keywords with MeSH terms (tw) for more focused results. To improve transparency, could you include the search strategy used for each database in the supplementary files? This would be helpful, including the initial search terms and the final results after removing duplicates.

Our search strategy includes both keywords and MeSH terms. We searched the database Medline using the Ovid interface, where MeSH terms are determined by a '/'. The MeSH terms used in this search can be found on lines 1-3, 6, 8-9, 14-15 and 21 in our Medline search strategy.

We have included an updated version of the search strategy (the file has been renamed as 'S2 Search Strategy') which includes the search terms for Embase and CINAHL and Web of Science. We were not able to include the search term for PsycInfo as the search was conducted by our Knowledge Synthesis & Medical Librarian at the University of British Columbia (Vanessa Kitchin, MI), who is currently on maternity leave until April 2025. We have requested access to this file from UBC Library services.

4. To enhance transparency in line with the PRISMA 2020 Checklist, could you consider including a table in the supplementary files that details the 12 excluded studies after full-text screening? The table could list each study and the corresponding reason for exclusion. This would provide valuable insight into the selection process.

We have included an additional file, 'S3 Reasons for Excluding Studies' with a table listing each study and the corresponding reason for exclusion.

5. What is the tool you used in your data extraction, analysis and synthesis? Could you please present and discuss the specific tool and present the data in your paper? Using certain tools for each selected study design can ensure the quality of evidence and reduce the risk of bias in the conduct of your study.

As described in our original submission, we used Covidence to manage independent title/abstract screening, full text screening, data extraction, and risk of bias assessment. We used Excel for our data charting tables to manage data extraction, analysis, and synthesis. We have clarified this under “Screening, study selection, and data charting” on lines 188-190.

‘After two co-authors (BO, ZK) conducted the searches, all identified citations were included in Covidence, a web-based software developed by the Cochrane group to streamline evidence synthesis reviews[20]. We used Covidence to manage independent title/abstract screening, full text screening, data extraction, and risk of bias assessment, after which we used Excel spreadsheets to manage data extraction, analysis, and synthesis.’

The JBI guidelines suggest the following regarding data extraction,

'Extraction of results for a scoping review should include extraction of all data relevant to inform the scoping review objective/s and question/s. Charting tables or forms may be used (see Appendix 10.1 for a template tool). A descriptive summary of the main results organized based on the review inclusion criteria must be included.'

Following this guidance, we adapted the JBI data charting table and used this table to extract relevant information from the selected studies. We previously described our data extraction process in the manuscript (now lines 199 to 204),

'We adapted relevant headings for our data charting table from the JBI guidelines and a scoping review on adolescent health [12,21]. The data charting table consisted of the following headings: study title, authors, study objectives, methods, context or setting, participant characteristics, data analysis, and author’s conclusions. Data extraction of the included studies was completed by one team member (VP), while a second member (BO) reviewed the extractions, consistent with the JBI scoping review guidelines.'

Similarly, we followed JBI guidelines to conduct a descriptive qualitative synthesis of the included studies. This process included reviewing the 'included studies and data charts to summarize the qualitative methods, characteristics of participants, and descriptions of youth engagement in the research process'. We have described our analysis in lines 206 to 213 of the manuscript, under the heading 'Analysis and synthesis'. We did not employe any specific 'tool' in the analysis of the qualitative data.

6. The manuscript mentions a data charting table with specific headings outlined on lines 160-162, including study title, authors, objectives, methods, context, participant characteristics, data analysis, and authors' conclusions. However, the paper currently only presents a table listing the included studies. For improved transparency and to allow readers to fully grasp the data analysis process, including the data charting table or a similar table with the mentioned headings in the supplementary files would be beneficial. This would provide a more comprehensive picture of the data extraction and analysis.

We have included an additional file, 'S4 Data Charting Table' with the requested information.

7. To improve the flow of information, consider combining the "Data" heading and its context onto the same page as the actual data content. This will prevent unnecessary page breaks and enhance readability. For optimal formatting, it's recommended to follow the specific guidelines outlined in the PLOS One criteria for the "Data" section. This will ensure a consistent and professional presentation of your data.

We do not have a heading named "Data" in the submitted manuscript. We require clarification on the reviewer's suggestion to combine the "Data" heading and its context onto the same page.

Response to Reviewer 1's Comments

We have made the following minor changes and tracked them on the updated manuscript:

27 Embase, CINAHL, PsycINFO). Of 415 identified [insert 'studies'] through the search process, five [delete 'studies'] were

28 included in this scoping review following screening for eligibility. We [replace 'report' with 'reported'] the findings of our

33 seeking general health care. We also [replace 'provide' with 'provided'] insights on formal and peer-based educational

34 interventions acceptable to newcomer youth. [replace 'Literature' with 'Literatures'] that qualitatively [replace 'describes' with 'describe'] newcomer

35 youth's experiences with contraception [replace 'is' with 'are'] scarce, and existing [replace 'literature' with 'literatur

---

## [Decision Letter · Decision Letter 1]

22 May 2025

Dear Dr. Munro,

Thank you for submitting your manuscript to PLOS ONE. After careful consideration, we feel that it has merit but does not fully meet PLOS ONE’s publication criteria as it currently stands. Therefore, we invite you to submit a revised version of the manuscript that addresses the points raised during the review process.

We look forward to receiving your revised manuscript.

Kind regards,

Funmilola M. OlaOlorun, PhD

Academic Editor

PLOS ONE

Journal Requirements:

[ZK is supported by the Affiliated Scholarship at the University of British Columbia, Mitacs Fellowship, the Systems Change Grant (2021) from the Vancouver Foundation and the Community and University Engagement Support Funds (2021) from the University of British Columbia. SM is supported by a Michael Smith Health Research BC Scholar Award (18270)].

[ZK is a board member of Options for Sexual Health (charity) and the Director of Free Periods Canada (registered non-profit society). ].

5. We noted in your submission details that a portion of your manuscript may have been presented or published elsewhere. [A version of this manuscript has been published as a chapter in a Master's thesis.] Please clarify whether this publication was peer-reviewed and formally published. If this work was previously peer-reviewed and published, in the cover letter please provide the reason that this work does not constitute dual publication and should be included in the current manuscript.

6. We note that your Data Availability Statement is currently as follows: [All relevant data are within the manuscript.]

The values behind the means, standard deviations and other measures reported;

The values used to build graphs;

The points extracted from images for analysis.

Additional Editor Comments (if provided):

Reviewers' comments:

Reviewer's Responses to Questions

**Comments to the Author**

Reviewer #3: All comments have been addressed

Reviewer #4: All comments have been addressed

2. Is the manuscript technically sound, and do the data support the conclusions?

Reviewer #3: Yes

Reviewer #4: Yes

3. Has the statistical analysis been performed appropriately and rigorously?

Reviewer #3: N/A

Reviewer #4: Yes

4. Have the authors made all data underlying the findings in their manuscript fully available?

Reviewer #3: Yes

Reviewer #4: Yes

5. Is the manuscript presented in an intelligible fashion and written in standard English?

Reviewer #3: Yes

Reviewer #4: Yes

Reviewer #3: This is an excellent contribution. I have only two very minor suggestions:

1. Page 3- perhaps clarify that although BC introduced free contraception in April 2023, people without a health card were still excluded?

2. Page 13- The Ashdown article is described as having participants "from Winnipeg", perhaps clarify the participants lived in Winnipeg but were newcomers? Then on page 19 the article is described as not having a disclosed Canadian city?

3. Excellent to describe the involvement of youth in the research process (where applicable). No suggestion, just praise.

Reviewer #4: The manuscript is technically sound with data supporting the findings, conclusions and recommendations.

**Do you want your identity to be public for this peer review?** For information about this choice, including consent withdrawal, please see our Privacy Policy

Reviewer #3: **Yes: ** Martha Paynter

Reviewer #4: **Yes: ** ALEXANDER LAAR

---

## [Author Response · Author response to Decision Letter 2]

6 Jun 2025

We thank the editor and peer reviewers for their helpful feedback and enthusiasm for this study. We feel the minor revisions have improved the manuscript and hope that the editors find it suitable for publication.

```````````````````````````````````````````

Responses to ‘Edits Requested’ June 3, 2025

We've checked your submission and before we can proceed, we need you to address the following issues:

1. We note that the grant information you provided in the ‘Funding Information’ and ‘Financial Disclosure’ sections do not match. When you resubmit, please ensure that you provide the correct grant numbers for the awards you received for your study in the ‘Funding Information’ section.

The correct grant numbers are included in the Funding Information section. We anticipate this comment refers to the two instances where “2021” is indicated in the Financial Disclosure (cited in comment 2 below). This refers to the year the funding was disbursed and is not an award number. We received different grants from each funder in subsequent years. The Funding Sources List in the Funding Information section references the correct Award Numbers for each grant.

[ZK is supported by the Affiliated Scholarship at the University of British Columbia, Mitacs Fellowship, the Systems Change Grant (2021) from the Vancouver Foundation and the Community and University Engagement Support Funds (2021) from the University of British Columbia. SM is supported by a Michael Smith Health Research BC Scholar Award (18270)].

The funders had no role in study design, data collection and analysis, decision to publish, or preparation of the manuscript. We have amended this in our cover letter. Thank you for changing the online submission form on our behalf.

[ZK is a board member of Options for Sexual Health (charity) and the Director of Free Periods Canada (registered non-profit society). ].

Please confirm that this does not alter your adherence to all PLOS ONE policies on sharing data and materials, by including the following statement: ""This does not alter our adherence to PLOS ONE policies on sharing data and materials.” (as detailed online in our guide for authors https://urldefense.com/v3/__http://journals.plos.org/plosone/s/competing-interests__;!!K-Hz7m0Vt54!imAqNsL-GAiyy02kDzf4k_fEf0kYWHL5VPrtzfVjuR1JWqK1C66pO2WbDP_CTQaK--UJ_lyG7hxF81s$ ). If there are restrictions on sharing of data and/or materials, please state these. Please note that we cannot proceed with consideration of your article until this information has been declared.

Co-author ZK’s competing interests do not alter our adherence to PLOS ONE policies on sharing data and materials. We have amended this in our cover letter. Thank you for changing the online submission form on our behalf.

4. We note that your Data Availability Statement is currently as follows: [All relevant data are within the manuscript.]

Please confirm at this time whether or not your submission contains all raw data required to replicate the results of your study. Authors must share the “minimal data set” for their submission. PLOS defines the minimal data set to consist of the data required to replicate all study findings reported in the article, as well as related metadata and methods (https://urldefense.com/v3/__https://journals.plos.org/plosone/s/data-availability*loc-minimal-data-set-definition__;Iw!!K-Hz7m0Vt54!imAqNsL-GAiyy02kDzf4k_fEf0kYWHL5VPrtzfVjuR1JWqK1C66pO2WbDP_CTQaK--UJ_lyG1TukIsw$ ).

The values behind the means, standard deviations and other measures reported;

The values used to build graphs;

The points extracted from images for analysis.

If your submission does not contain these data, please either upload them as Supporting Information files or deposit them to a stable, public repository and provide us with the relevant URLs, DOIs, or accession numbers. For a list of recommended repositories, please see https://urldefense.com/v3/__https://journals.plos.org/plosone/s/recommended-repositories__;!!K-Hz7m0Vt54!imAqNsL-GAiyy02kDzf4k_fEf0kYWHL5VPrtzfVjuR1JWqK1C66pO2WbDP_CTQaK--UJ_lyGsObF5cI$ .

All relevant data and methods are within the manuscript and the Supplementary Files. This is a scoping review of qualitative literature. There are no quantitative means, standard deviations, other measures, graphs, etc. in this manuscript. The search strategy and data extraction methods and forms are included in our files, making it possible for anyone to replicate the results of our study.

5. Please include captions for your Supporting Information files at the end of your manuscript, and update any in-text citations to match accordingly. Please see our Supporting Information guidelines for more information: https://urldefense.com/v3/__http://journals.plos.org/plosone/s/supporting-information__;!!K-Hz7m0Vt54!imAqNsL-GAiyy02kDzf4k_fEf0kYWHL5VPrtzfVjuR1JWqK1C66pO2WbDP_CTQaK--UJ_lyG0rp8e1w$ .

We have added an intext caption for our S3 supporting information files. All others were included at the end of the manuscript (page 8) and referenced in text following journal guidelines:

• S1. PRISMA-ScR Checklist (page 5, line 104)

• S2. Search Strategy (page 6, line 120)

• S3. Reasons for Excluding Studies (added to page 8, line 153)

• S4. Data Charting Table (cited on page 10, line 178)

6. Please ensure that you refer to Tables 1 and 2 in your text as, if accepted, production will need this reference to link the reader to the Table.

The reference to Tables 1 is in text (p. 8, line 160) and we have added a reference to Table 2 (page 11, line 195).

````````````````````````````

Responses to ‘Comment’ May 22, 2025

1. The result of this report is published at the University of Columbia as a thesis report. So, it is not original report.

We would like to clarify that a version of this manuscript was a chapter in a master’s thesis, which was archived in an online repository. This work has not been published in a peer-reviewed journal.

2. Key words should be related to the research title e.g. Emigrants

We included the following keywords in the manuscript: ‘Canada, Contraception, Emigrants and Immigrants, Health Services Accessibility’. These keywords are related to the title of our manuscript, ‘Newcomer youth’s access to contraception care in Canada: A scoping review of qualitative evidence’.

3. Youth refers to individuals aged 15 to 25 years. What is your source/reference?

We revised this sentence to, ‘In this review, youth refers to individuals aged 15 to 25 years.’ Appropriate references are included in lines 147-149.

4. Your study participants are between aged 15-25 years. You include the study conducted in Ghana by Boafo- Arthur (2013) which conducted on participants between aged 18-27 years which has difference in age. So, why you include? And how can draw conclusion from it? Clearly state it.

The term ‘youth’ refer to a period when individuals transition from childhood to adulthood. The categorization of youth is fluid, and context dependent. There are no universally agreed upon age ranges for ‘youth’. Although some participants in that study fall outside our specified age range, the overlap in key themes related to youth SRH justify its inclusion.

5. Another gap I will found in this study was current literature on healthcare access for immigrant populations identify language as a critical barrier for newcomers who do not speak English as their first language [27]. The results of our scoping review provide additional insights into the language preferences of youth specifically regarding SRH topics. Newcomer youth who spoke English as a second language expressed a preference for discussing SRH topics in English. How?

While some literature note language as a barrier, our review highlights a more nuanced perspective from one study (Ashdown et.al 2015), in which newcomer youth described their preferences for using certain SRH terms in English.

We address this in lines 321-323, ‘Youth participants further elaborated that the language used to communicate topics related to SRH had cultural implications beyond comprehension [23]. For example, some newcomer participants shared that words like penis, oral, and sex are not widely used in their first language, and expressed feeling more comfortable discussing these terms and related topics in English [23].’

6. Conclusion should be based on your research question.

Thank you for this comment. The purpose of our scoping review was to ‘summarize the current qualitative evidence of newcomer youths’ experiences with accessing contraception care and highlight gaps in the evidence.’ In the conclusion of our manuscript, we reiterate our key findings in relation to this objective, including the identified evidence gaps. We believe the conclusion is aligned with our stated research purpose.

7. Technically the way they are planning for the teams are good for the contribution of the knowledge and skills in the study to assure the soundness of the finding.

Noted.

Responses to #Reviewer 3

1. Page 3- perhaps clarify that although BC introduced free contraception in April 2023, people without a health card were still excluded?

Thank you for the feedback. We amended lines 74-75 to address this comment.

2. Page 13- The Ashdown article is described as having participants "from Winnipeg", perhaps clarify the participants lived in Winnipeg but were newcomers? Then on page 19 the article is described as not having a disclosed Canadian city?

Thank you for drawing our attention to this discrepancy. Upon further review, we confirm that the authors did not explicitly state the name of the city where the study took place, rather we abstracted this from the information provided. We removed ‘Winnipeg, Manitoba’ from Table 2, and replaced it with ‘Participants were from a Canadian city (the name of the city was not disclosed in the study).

The authors are affiliated with institutions in Winnipeg, Manitoba, and they refer to a separate study conducted in Winnipeg in the article.

3. Excellent to describe the involvement of youth in the research process (where applicable). No suggestion, just praise.

We appreciate this reflection from the reviewer.

Journal Requirements

We have amended the formatting of the names to meet PLOS One’s style requirements.

[ZK is supported by the Affiliated Scholarship at the University of British Columbia, Mitacs Fellowship, the Systems Change Grant (2021) from the Vancouver Foundation and the Community and University Engagement Support Funds (2021) from the University of British Columbia. SM is supported by a Michael Smith Health Research BC Scholar Award (18270)].

Please amend the financial disclosure to the following:

ZK is supported by the Affiliated Scholarship at the University of British Columbia, Mitacs Fellowship, the Systems Change Grant (2021) from the Vancouver Foundation and the Community and University Engagement Support Funds (2021) from the University of British Columbia. SM is supported by a Michael Smith Health Research BC Scholar Award (18270). The funders had no role in study design, data collection and analysis, decision to publish, or preparation of the manuscript.

[ZK is a board member of Options for Sexual Health (charity) and the Director of Free Periods Canada (registered non-profit society). ].

Please amend the competing interests statement to the following: ‘ZK is a board member of Options for Sexual Health (charity) and the Director of Free Periods Canada (registered non-profit society). This does not alter our adherence to PLOS ONE policies on sharing data and materials.

5. We noted in your submission details that a portion of your manuscript may have been presented or published elsewhere. [A version of this manuscript has been published as a chapter in a Master's thesis.] Please clarify whether this publication was peer-reviewed and formally published. If this work was previously peer-reviewed and published, in the cover letter please provide the reason that this work does not constitute dual publication and should be included in the current manuscript.

We would like to clarify that a version of this manuscript was a chapter in a master’s thesis, which has been archived in an online repository. This work has not been published in a peer-reviewed journal, and therefore, the current submission would not constitute dual publication.

6. We note that your Data Availability Statement is currently as follows: [All relevant data are within the manuscript.]

The values

---

## [Editor Report · Decision Letter 2]

25 Jun 2025

Newcomer youth’s access to contraception care in Canada: A scoping review of qualitative evidence

PONE-D-24-05128R2

Dear Dr. Munro,

We’re pleased to inform you that your manuscript has been judged scientifically suitable for publication and will be formally accepted for publication once it meets all outstanding technical requirements.

Kind regards,

Funmilola M. OlaOlorun, PhD

Academic Editor

PLOS ONE
---

## [Editor Report · Acceptance letter]

PONE-D-24-05128R2

PLOS ONE

Dear Dr. Munro,

I'm pleased to inform you that your manuscript has been deemed suitable for publication in PLOS ONE. Congratulations! Your manuscript is now being handed over to our production team.

Kind regards,

on behalf of

Dr. Funmilola M. OlaOlorun

Academic Editor

PLOS ONE